# OpenReview forum: "Self-Detoxifying Language Models via Toxification Reversal"
_EMNLP/2023/Conference — EMNLP 2023 Main_

### Official Review · Reviewer_tv99 · 2023-07-26

**Soundness:** 4

**Excitement:**

4: Strong: This paper deepens the understanding of some phenomenon or lowers the barriers to an existing research direction.

**Paper Topic And Main Contributions:**

In this work, the authors propose a new prompt-based method for language detoxification. In each step during inference, two forward passes are needed. In the first pass, a negative prompt and a positive prompt are prepended to the input respectively and a toxification direction is calculated. In the second pass, the contextualized value vector corresponding to the last token is steered away from the toxification direction. Experimental results show that the method achieves significant improvement over prompt-based baselines and outperforms a few finetuning-based and decoding-based baselines.


**Questions For The Authors:**

1. Would you provide more disscussions on computational cost?
2. Will you conduct experiments on more base models?

**Reasons To Accept:**

1. The method is simple (in a good way) and effective.
2. The experiment design is solid.

**Reasons To Reject:**

1. Significant extra computational cost is introduced.
2. The experiments are only conducted on GPT-2.

**Reproducibility:**

5: Could easily reproduce the results.

**Reviewer Confidence:**

2: Willing to defend my evaluation, but it is fairly likely that I missed some details, didn't understand some central points, or can't be sure about the novelty of the work.

---

> ### Author Rebuttal · Authors · 2023-08-26
>
> We greatly appreciate your thoughtful review and feedback on our paper.  We will address your concerns below.
>
> **Q1: "Significant extra computational cost is introduced. Would you provide more discussions on computational cost?"**
>
> A1:  We'd like to clarify that our method, while introducing additional computational steps, does not significantly increase the computational cost compared to other methods.
>
> Firstly, in comparison to finetuning-based models, our method does not require any additional training. The fine-tuning process for language models (LM) is computationally demanding, especially given the increasing size of LMs. Our method, conversely, eliminates this need, reducing the computational load.
>
> Secondly, when compared to decoding-based methods, our model does not incorporate any extra modules. The table below illustrates that two decoding-based baselines introduce additional parameters to the base model, while ours does not. Consequently, our method's basic memory requirements are less than these alternative approaches.
>
> | Method | DEXPERTS | GeDi  | **Ours** |
> | ------ | -------- | ----- | -------- |
> | Param  | 2322M    | 1129M | **774M** |
>
> Thirdly, our method demonstrates superior inference latency compared to the sota decoding-based method, as presented in the following table. We derived these results from 100 randomly sampled prompts in the dataset with a batch size of 1, conducted on a 3090 GPU.
>
> | Method                 | DEXPERTS | Ours     | **Ours***    |
> | ---------------------- | -------- | -------- | ------------ |
> | Latency(ms±std/sample) | 943±12ms | 828±14ms | **756±14ms** |
>
> ("Ours*" means not doing toxification reversal in the bottom 16 layers, since this only leads to a marginal detoxification performance decay, as discussed in section 5.1.1.)
>
> When combined with the results in Table 3, our method provides a competitive performance without significantly increasing computational costs. We agree that it would be beneficial to have a more comprehensive discussion on computational cost in our paper. We will address this in the appendix.
>
> **Q2: "The experiments are only conducted on GPT-2. Will you conduct experiments on more base models?"**
>
> A2: We choose GPT-2 as the backbone of our method for evaluation in order to be consistent with the practice in previous works in this area. Nonetheless, we acknowledge the value of diversifying the base models used in the experiments.  We will actively consider including additional base models, such as llama, in our next revision.

---

### Official Review · Reviewer_rKEz · 2023-08-04

**Soundness:** 4

**Excitement:**

4: Strong: This paper deepens the understanding of some phenomenon or lowers the barriers to an existing research direction.

**Missing References:**

David Dale, Anton Voronov, Daryna Dementieva, Varvara Logacheva, Olga Kozlova, Nikita Semenov, and Alexander Panchenko. 2021. Text Detoxification using Large Pre-trained Neural Models. In Proceedings of the 2021 Conference on Empirical Methods in Natural Language Processing, pages 7979–7996, Online and Punta Cana, Dominican Republic. Association for Computational Linguistics.

Varvara Logacheva, Daryna Dementieva, Sergey Ustyantsev, Daniil Moskovskiy, David Dale, Irina Krotova, Nikita Semenov, and Alexander Panchenko. 2022. ParaDetox: Detoxification with Parallel Data. In Proceedings of the 60th Annual Meeting of the Association for Computational Linguistics (Volume 1: Long Papers), pages 6804–6818, Dublin, Ireland. Association for Computational Linguistics.

**Paper Topic And Main Contributions:**

The paper presents a new method to stop the model to generate toxic outputs but generate non-toxic ones. This method is based on the idea of two "trajectories" -- toxification and toxification reversal, detoxification.

**Questions For The Authors:**

Questions A: how do you formulate the task of LLM detoxification -- generate the same output but non-toxic or generate something non-toxic?

Question B: Figure 2: Negative vs Positive prefix == toxic, negative text vs polite, positive text. To detoxify text does not mean to make it polite or positive. The negative (or any other) sentiment should be saved. We just want to generate neutral text. That is why more precise definition of what your goal is required.

Question C: Table 1: what do underline or bold mean, what is the difference? Which metrics should be compared as the bigger the better, which once the less the better?

Questions D: if we compare the proposed method with output text detoxification results, what do you think might be better?

Question E: what is the reason, in your opinion, of such low agreement between annotators?

**Reasons To Accept:**

- The new method for controllable text generation is presented;
- The evaluation setup design is thorough;
- Both automatic and manual evaluations are present.

**Reasons To Reject:**

- No comparison with text detoxification baselines. There are already seq2seq approaches for text detoxification (https://huggingface.co/s-nlp/bart-base-detox). There should be a justification that the generated output is indeed better than just detoxification of toxic output. I understand, that this approach will require additional inference, however, in terms of generate text quality, we do not know for now if it is better or worse.
- Only one model is used to text generation -- GPT-2.
- After looking at the examples, for me, it is unclear what do we want to achieve with LLMs detoxification -- do we want to generate the same output but non-toxic or we just want to generate something non-toxic? The paper lacks of more precise task definition and discussion in this regard.
- Continuing previous issue, it is possible that additional evaluation in terms of content similarity or at least content relevance is needed.
- Annotator agreement in human evaluation is low.

**Reproducibility:**

4: Could mostly reproduce the results, but there may be some variation because of sample variance or minor variations in their interpretation of the protocol or method.

**Reviewer Confidence:**

4: Quite sure. I tried to check the important points carefully. It's unlikely, though conceivable, that I missed something that should affect my ratings.

**Typos Grammar Style And Presentation Improvements:**

Table 1: describe what is the difference between bold and underline numbers, how the scores should be compared.

Figure 2: please, revise what do you mean by positive prefix.

---

> ### Author Rebuttal · Authors · 2023-08-26
>
> Thank you for your comments and feedback on our submission.  We will address your concerns and clarify any misunderstandings below.
>
> **Q1: "how do you formulate the task of LLM detoxification -- generate the same output but non-toxic or generate something non-toxic?"**
>
> A1: In the first place, we want to point out that reviewer rKEz might have some misunderstanding about our task definition for language detoxification. Our task definition is in line with the previous paper listed in the related works. As explained in lines 110-120, we formulate this task as making LM generate non-toxic texts. It needs to be clarified that our focused task is different from the text detoxification task that intends to detoxify an existing text. We aim to develop methods to avoid toxic generation from pretrained LMs, while the latter develops methods to convert a given toxic text into a non-toxic one.
>
> **Q2: "Figure 2: Negative vs Positive prefix == toxic, negative text vs polite, positive text. To detoxify text does not mean to make it polite or positive. The negative (or any other) sentiment should be saved. We just want to generate neutral text. That is why more precise definition of what your goal is required." and “Figure 2: please, revise what do you mean by positive prefix.”**
>
> A2: Positive/negative prefix does not indicate encouragement for positive/negative sentiment. As we point out in A1, the goal is to reduce the risk of generating toxic content (e.g., rude, disrespectful, or insulting content that hinders AI safety), but not restrict the model to only generate polite or positive content.  As we demonstrate in Appendix.D, the positive prefix serves as a contrasting reference for a better toxification direction discovery, minimizing the negative effects on context representation. This is proved in Table 2 in Appendix.D, where using the positive prefix yields better perplexity with similar toxicity reduction scores. We will add more discussion about the prefixes in the method section.
>
> **Q3:  "Table 1: describe what is the difference between bold and underline numbers, how the scores should be compared."**
>
> A3: The Exp. Max. Tox. (Expected Maximum Toxicity) and Tox. Prob. (Toxicity Probability) are used to measure how toxic the generated texts are; thus, the lower, the better. The PPL (Perplexity) is used to measure how fluent the generated texts are, and the lower, the better. \textcolor{red}{Therefore,} we use bold numbers to indicate the best result among Prompt-based methods, where our method lies, for a fair comparison. We underline the lowest scores among all methods for clarity. We will add more illustrations for the clarity of the results.
>
> **Q4: "if we compare the proposed method with output text detoxification results, what do you think might be better?"**
>
> A4: As we point out in A1, our primary focus lies in steering the LM to naturally produce non-toxic content, but not in retargeting the same message in a non-toxic manner. Therefore,  comparing our method directly with text detoxification methods may not be appropriate, as the two approaches aim to solve different aspects. We will add discussion and distinction of text detoxification methods in the related works.
>
> **Q5: "what is the reason, in your opinion, of such low agreement between annotators?"**
>
> A5: Our Fleiss's Kappa value falls within the range of 'fair agreement (0.21 <= $\kappa$ <= 0.40)' according to the established scale (Landis et al. 1977). This suggests a moderate degree of consensus among the evaluators, despite the evaluation's challenging and subjective nature. The assessment of language toxicity involves a nuanced understanding of the language. Moreover, different academic and cultural backgrounds, along with individual experiences and biases, could have contributed to this divergence in opinions. Therefore,  it is inevitable to have relatively low agreement between the annotators for this task.
>
> **Q6: "Only one model is used to text generation -- GPT-2."**
>
> A6: The choice to conduct experiments solely on GPT-2 was guided by its widespread acceptance and use in related works. However, we acknowledge the value of diversifying the base models used in the experiments.  We will actively consider including additional base models, such as llama, in our next revision.
>
>
>
> Landis, J. R., & Koch, G. G. (1977). The Measurement of Observer Agreement for Categorical Data. *Biometrics*, *33*(1), 159–174.

---

### Official Review · Reviewer_fNXE · 2023-08-10

**Soundness:** 4

**Excitement:**

4: Strong: This paper deepens the understanding of some phenomenon or lowers the barriers to an existing research direction.

**Paper Topic And Main Contributions:**

### Self-Detoxifying Language Models via Toxification Reversal

The authors propose a lightweight approach to detoxify pretrained large language models. The main idea is to feed toxic samples into a generative model and measure the direction of the generated text in the latent vector space. This direction is the toxic direction. To detoxify a model's output, it is steered in the opposite direction.
The authors compare their approach to existing detoxification methods, apply automatic and human evaluation, and find that the proposed method can efficiently reduce toxicity.

**Questions For The Authors:**

Will the implementation be released?

**Reasons To Accept:**

The paper is well-written and easy to follow. The contribution and its relevance are clear. The methodology is mathematically well-substantiated and well-evaluated.
Generating latent steering vectors to affect a model's output is a state-of-the-art concept. Although the achieved detoxification relies largely on the selected data samples, the present implementation is a valuable and novel contribution to that idea. I believe that this is a good paper and should be accepted, as it addresses ongoing topics.

**Reasons To Reject:**

The human evaluation step seems to be rather a supplementary layer to the formal evaluation steps. Nevertheless, three evaluators are few for the strong conclusion. I assume each of these has rated 50 samples, not all 150 samples. That is not entirely clear.
"with related backgrounds" means they are graduate students in NLP? It could be even more meaningful to recruit neutral evaluators to show the general validity of the results.

**Reproducibility:**

3: Could reproduce the results with some difficulty. The settings of parameters are underspecified or subjectively determined; the training/evaluation data are not widely available.

**Reviewer Confidence:**

4: Quite sure. I tried to check the important points carefully. It's unlikely, though conceivable, that I missed something that should affect my ratings.

**Typos Grammar Style And Presentation Improvements:**

The approach is pretty much based on individual samples. I would like to see a few examples. For instance, those that were presented to the evaluators.

---

> ### Author Rebuttal · Authors · 2023-08-26
>
> We appreciate your thoughtful comments and suggestions, and we are glad to learn that you found our contribution valuable to the field. We address your specific concerns and questions below.
>
> **Q1: three evaluators are few for the strong conclusion, and I assume each of these has rated 50 samples, not all 150 samples.**
>
> A1:
>
> - In line with the approach adopted by Schick et al. (2021), we engaged three professional annotators to conduct our human evaluation. Considering this as a common practice in this task, we maintain the same number of annotators for our evaluation process.
> - In order to calculate the evaluation consistency, we ask each evaluator to conduct all 150 comparisons in three different aspects (i.e., 50 for ''Ours vs. DAPT'', 50 for ''Ours vs. DExperts'', and 50 for ''Ours vs. SD''). We will revise the descriptions of the human evaluation setting to make it clearer.
>
> **Q2:  "with related backgrounds" means they are graduate students in NLP? It could be more meaningful to recruit neutral evaluators to show the validity of the results.**
>
> A2: All three human annotators involved in our evaluation are graduate students with NLP background.  However, their academic focus doesn't align with the research area of this study. Moreover, they have not participated in our research process or contributed to the development of the tested model, ensuring their neutrality.
>
> **Q3: Will the implementation be released?**
>
> A3: We will make our codes available upon acceptance.
>
>
>
>
> Schick, T., Udupa, S., & Schütze, H. (2021). Self-Diagnosis and Self-Debiasing: A Proposal for Reducing Corpus-Based Bias in NLP. Transactions of the Association for Computational Linguistics, 9, 1408–1424.

---

### Meta-Review · Area_Chair_Vy3q · 2023-09-19

**Recommendation:** 4

**Metareview:**

The paper introduces a method for detoxifying language models by identifying and reversing toxification trajectories in the latent vector space. The main premise is to leverage toxic samples in a generative model to determine the toxic direction. By moving in the opposite direction, the model's output can be detoxified. This method's efficacy is gauged against other detoxification methods through both automatic and human evaluation. For most reviewers, the paper presents an innovative approach to language detoxification with a well-defined mathematical foundation. It introduces a potentially groundbreaking method, However, there are concerns about the clarity of its objectives, the evaluation process, computational costs, and the limitation to GPT-2. These factors should be addressed for a more comprehensive contribution. A reviewer expressed appreciation for the effort and dedication in providing experimental results. They found clarity in the response provided, specifically noting the differences between toxic and detoxified continuations from Table 3. While both content types were coherent, This reviewer raised concerns regarding the use of perplexity as a measure due to its observed low correlation with manual assessments in many generation tasks. However, they acknowledged that this might be a broader scientific discussion rather than a direct critique of the presented work. In conclusion, the reviewer encouraged the addition of two-step pipeline experiments and more prompt examples (including from other baseline models) to enhance comprehension for both annotators and readers.
Another reviewer acknowledges the author's clarifications but raises further concerns about the fundamental understanding of the task. While both language detoxification (preventing toxic language generation) and text detoxification (reducing toxicity of existing text) aim for a non-toxic output, they operate differently. The central question is whether detoxifying the "language" from which tokens are sampled affects the relevance of the generated text compared to the original Language Model's output. The reviewer posits that perhaps only certain toxic words need detoxifying while retaining content relevance. They believe that the evaluation methodology for this task might not be clearly established.

---

### Decision · Program_Chairs · 2023-10-07

**Decision:**

Accept-Main

**Comment:**

The paper introduces a method for detoxifying language models by identifying and reversing toxification trajectories in the latent vector space. The main premise is to leverage toxic samples in a generative model to determine the toxic direction. By moving in the opposite direction, the model's output can be detoxified. This method's efficacy is gauged against other detoxification methods through both automatic and human evaluation. For most reviewers, the paper presents an innovative approach to language detoxification with a well-defined mathematical foundation. It introduces a potentially groundbreaking method, However, there are concerns about the clarity of its objectives, the evaluation process, computational costs, and the limitation to GPT-2. These factors should be addressed for a more comprehensive contribution. A reviewer expressed appreciation for the effort and dedication in providing experimental results. They found clarity in the response provided, specifically noting the differences between toxic and detoxified continuations from Table 3. While both content types were coherent, This reviewer raised concerns regarding the use of perplexity as a measure due to its observed low correlation with manual assessments in many generation tasks. However, they acknowledged that this might be a broader scientific discussion rather than a direct critique of the presented work. In conclusion, the reviewer encouraged the addition of two-step pipeline experiments and more prompt examples (including from other baseline models) to enhance comprehension for both annotators and readers.
Another reviewer acknowledges the author's clarifications but raises further concerns about the fundamental understanding of the task. While both language detoxification (preventing toxic language generation) and text detoxification (reducing toxicity of existing text) aim for a non-toxic output, they operate differently. The central question is whether detoxifying the "language" from which tokens are sampled affects the relevance of the generated text compared to the original Language Model's output. The reviewer posits that perhaps only certain toxic words need detoxifying while retaining content relevance. They believe that the evaluation methodology for this task might not be clearly established.